healthcareCOVID: a national cross-sectional observational study identifying risk factors for developing suspected or confirmed COVID-19 in UK healthcare workers

http://orcid.org/0000-0002-5741-2300 Kua Justin 1 2 justin.kua.20@ucl.ac.uk
Patel Reshma 1 2
Nurmi Eveliina 3
Tian Sarah 4
Gill Harpreet 4
http://orcid.org/0000-0002-3524-1766 Wong Danny J.N. 2 4
http://orcid.org/0000-0002-5969-2465 Moorley Calvin 5
http://orcid.org/0000-0002-2171-2862 Nepogodiev Dmitri 6
Ahmad Imran 4 7
http://orcid.org/0000-0002-9912-717X El-Boghdadly Kariem 4 7
1 Department of Targeted Intervention, Centre for Perioperative Medicine, UCL Division of Surgery and Interventional Science, University College London, University of London , London , United Kingdom
2 Health Services Research Centre, National Institute of Academic Anaesthesia, Royal College of Anaesthetists , London , UK
3 Department of Anaesthesia, University College London Hospitals NHS Foundation Trust , London , UK
4 Department of Anaesthesia, Guy’s and St Thomas’ NHS Foundation Trust , London , UK
5 School of Health and Social Care/Adult Nursing & Midwifery Studies, London South Bank University , London , UK
6 National Institute for Health Research Global Health Research Unit on Global Surgery, University of Birmingham , Birmingham , UK
7 King’s College London, University of London , London , UK
Khiabanian Hossein
Electronic publication date: 2021 Feb 4
Publication date: 2021
Volume: 9
Electronic Location ID: e10891
Received 2020 Oct 29; Accepted 2021 Jan 12
Copyright: © 2021 Kua et al.
Copyright year: 2021
Copyright holder: Kua et al.
License: This is an open access article distributed under the terms of the Creative Commons Attribution License, which permits unrestricted use, distribution, reproduction and adaptation in any medium and for any purpose provided that it is properly attributed. For attribution, the original author(s), title, publication source (PeerJ) and either DOI or URL of the article must be cited.
License URL: https://creativecommons.org/licenses/by/4.0/

Keywords: Coronavirus, COVID-19, SARS-CoV-2, Healthcare workers, Medical workers

Funding: The authors received no funding for this work.

==============================
Objective

To establish the prevalence, risk factors and implications of suspected or confirmed coronavirus disease 2019 (COVID-19) infection among healthcare workers in the United Kingdom (UK).

Design

Cross-sectional observational study.

Setting

UK-based primary and secondary care.

Participants

Healthcare workers aged ≥18 years working between 1 February and 25 May 2020.

Main outcome measures

A composite endpoint of laboratory-confirmed diagnosis of SARS-CoV-2, or self-isolation or hospitalisation due to suspected or confirmed COVID-19.

Results

Of 6,152 eligible responses, the composite endpoint was present in 1,806 (29.4%) healthcare workers, of whom 49 (0.8%) were hospitalised, 459 (7.5%) tested positive for SARS-CoV-2, and 1,776 (28.9%) reported self-isolation. Overall, between 11,870 and 21,158 days of self-isolation were required by the cohort, equalling approximately 71 to 127 working days lost per 1,000 working days. The strongest risk factor associated with the presence of the primary composite endpoint was increasing frequency of contact with suspected or confirmed COVID-19 cases without adequate personal protective equipment (PPE): ‘Never’ (reference), ‘Rarely’ (adjusted odds ratio 1.06, (95% confidence interval: [0.87–1.29])), ‘Sometimes’ (1.7 [1.37–2.10]), ‘Often’ (1.84 [1.28–2.63]), ‘Always’ (2.93, [1.75–5.06]). Additionally, several comorbidities (cancer, respiratory disease, and obesity); working in a ‘doctors’ role; using public transportation for work; regular contact with suspected or confirmed COVID-19 patients; and lack of PPE were also associated with the presence of the primary endpoint. A total of 1,382 (22.5%) healthcare workers reported lacking access to PPE items while having clinical contact with suspected or confirmed COVID-19 cases.

Conclusions

Suspected or confirmed COVID-19 was more common in healthcare workers than in the general population and is associated with significant workforce implications. Risk factors included inadequate PPE, which was reported by nearly a quarter of healthcare workers. Governments and policymakers must ensure adequate PPE is available as well as developing strategies to mitigate risk for high-risk healthcare workers during future COVID-19 waves.

Introduction

Coronavirus disease 2019 (COVID-19), caused by severe acute respiratory syndrome coronavirus 2 (SARS-CoV-2), has resulted in a global health crisis that has challenged healthcare systems around the world. As of 26 January 2021, there are nearly 100 million confirmed cases and more than 2 million deaths worldwide (World Health Organization, 2020a). Healthcare workers have been identified to be at risk of nosocomial COVID-19 infection (Kursumovic, Lennane & Cook, 2020; El-Boghdadly et al., 2020; Pollán et al., 2020; Shields et al., 2020). In the United States (US), over 379,000 healthcare workers have been infected with COVID-19 (CDC, 2020) and up to 16–17.2% of those infected in the UK were thought to be key workers, a category that includes both healthcare workers as well as other essential workers from other industries (Heneghan, Oke & Jefferson, 2020; Shah et al., 2020).

The prevalence of COVID-19 in healthcare workers is thought to be higher than in the general population, potentially due to exposure to higher viral loads from increased contact with infected individuals (Heneghan, Brassey & Jefferson, 2020; Wilson et al., 2020; El-Boghdadly et al., 2020). Prevalence estimates are variable with limited data comparing different staff roles and workplace environments (i.e. secondary versus primary care) (Office for National Statistics, 2020; Keeley et al., 2020; Hunter et al., 2020; Nguyen et al., 2020; Shields et al., 2020). This, combined with the limited availability of testing for healthcare workers during early phases of the pandemic, resulted in reliance on periods of self-isolation as a means of controlling the spread of the virus (Dunn et al., 2020). The impact of self-isolation on the UK healthcare workforce during the height of the pandemic has not been characterised.

Reports of healthcare workers deaths have also revealed that Black, Asian and Minority Ethnic (BAME) groups appear to represent a greater proportion of these deaths (Kursumovic, Lennane & Cook, 2020). The reasons for this preponderance of BAME groups and COVID-19 severity in healthcare workers as well as the general population are likely to be complex and multifactorial (Intensive Care National Audit & Research Centre, 2020; Aldridge et al., 2020; Kursumovic, Lennane & Cook, 2020). Similarly, healthcare worker-related mortality from COVID-19 in the UK is reported as one of the highest globally, yet the reasons for this are poorly understood (Amnesty International UK, 2020).

Given the potential for aerosol transmission of SARS-CoV-2, healthcare workers exposed to aerosol-generating procedures (AGPs) are at potentially higher risk of developing COVID-19 (Cook, 2020; Wilson et al., 2020; El-Boghdadly et al., 2020). However, what constitutes an AGP remains contentious, with conflicting international and national guidance (Public Health England, 2020c; World Health Organization, 2020b). Furthermore, shortages of personal protective equipment (PPE) throughout the pandemic and beyond remain a concern for healthcare workers (The Lancet, 2020). Taken together, the workplace environmental risks for healthcare workers with different exposures to COVID-19 and access to PPE remain unclear, particularly in the early surges of the pandemic.

We therefore designed a UK-wide cross-sectional study to understand the prevalence and possible risk factors for the reporting of suspected or confirmed COVID-19 infection amongst healthcare workers. We also sought to estimate access to testing and the number of working days lost from self-isolation. Finally, we aimed to capture details on socio-demographics, occupational exposure, and use of PPE to help expand the evidence base for healthcare workers and policymakers.

Materials and Methods

We conducted a cross-sectional observational study of UK-based healthcare workers between 4 and 25 May 2020 in England, Northern Ireland, Scotland and Wales. We included all healthcare workers aged 18 years or above and working at any time since 1 February 2020. Healthcare workers practising in both primary (community and social care facilities) as well as secondary (hospitals) care were eligible. The study was prospectively registered as a service evaluation project at Guy’s and St Thomas’ NHS Foundation Trust (Service Evaluation ID: 10834) and was deemed to not require ethical approval by the Research and Development Department and the Health Research Authority Decision Tool.

Study design

We designed an online survey using Knack (Evenly Odd Inc., Philadelphia, PA, USA), an online data capture and database system. A multi-phased process involving several authors (JK, RP, KE, IA, DW, CM) was used to construct, revise and ratify the final survey. An initial draft of questions for the survey was created by JK and RP and sent to the remaining authors for review. Based on feedback received, modifications were made and questions compiled, followed by a second round of review and testing by all authors on the online system. This version of the survey was piloted in a convenience sample of 93 participants. One change was made as a consequence: an expanded list of specialties was implemented.

The final survey comprised 33 closed questions and five free-text entries, divided into five sections: (1) participant characteristics; (2) work details; (3) self-isolation and COVID-19 status; (4) workplace exposure characteristics; and (5) PPE (Supplemental Material). Free-text entries were used for gender identity (if not the same as sex at birth), ethnic background (if not within one of the listed groups), number of days of self-isolation (if greater than 14 days, with conditional limits) and an ‘Other Comments’ question. The survey covered experiences from the period 1 February 2020 to the date when each healthcare worker participated in the study.

Survey administration

The survey was disseminated electronically using a web link which directed potential participants to the survey form. This web link was shared on several relevant social media platforms and via e-mail. We engaged several organisations and Royal Colleges to assist with dissemination to their respective membership, which included the Association for Perioperative Practice, the COVID-19 Information Hub for the Royal College of Surgeons of England, the Royal College of Occupational Therapists, and the Association of Anatomical Pathology Technology.

Definitions

Several survey response variables were grouped a priori to facilitate analyses. We defined a collective ‘BAME’ ethnic group as those participants who identified as ‘Asian or Asian British’, ‘Black, African, Black British or Caribbean’, ‘Mixed or multiple ethnic groups’, and ‘another ethnic group’, in keeping with contemporary reporting (Intensive Care National Audit & Research Centre, 2020; Aldridge et al., 2020; Kursumovic, Lennane & Cook, 2020). Occupational roles were grouped into five subgroups: (1) Doctors—all doctors; (2) Dentists and dental staff—dentist, dental nurse, and dental hygienist; (3) Nurses, midwives and associated staff—healthcare assistant, maternity care worker, midwife, nurse, and nursing associate; (4) Allied Health Professionals (AHPs)—dietician, healthcare scientist (e.g. lab-based), occupational therapist, operating department practitioner, optician, paramedic, pharmacist, phlebotomist, physician associate, physiotherapist, psychologist, radiographer, speech and language therapist, technician (clinical), and therapist (Other); and (5) Other—administrative staff, domestic services, manager (care home), ‘other’, porter, senior carer (care home), support worker/assistant, and wellbeing/activity coordinator (care home). In line with Public Health England (PHE) guidance, (Public Health England, 2020c) higher risk areas were considered to be the following: COVID-19 pod/bay/ward, day case surgery unit, emergency department (ED), endoscopy unit (upper respiratory, ENT or upper GI endoscopy), intensive care (ICU)/High dependency unit (HDU), and operating theatre.

We originally included an option for ‘Intersex’ when enquiring about sex and gender identity to support inclusivity based on published guidance (Reisner et al., 2014; Spiel, Haimson & Lottridge, 2019). However, during the study, several healthcare workers and members of the public expressed concern regarding this approach, leading to a removal of the option for ‘Intersex’, leaving only ‘Male’ and ‘Female’ as options for sexual identity. We retained the question about gender identity, including a free-text option for those who identified as a gender not the same as sex at birth.

The primary endpoint of this study was a composite outcome of any of the following: (1) self-isolation due to COVID-19 symptoms or a positive SARS-CoV-2 test, (2) hospitalisation with suspected or confirmed COVID-19 and (3) laboratory-confirmed SARS-CoV-2 infection (via reverse transcription polymerase chain reaction or antibody testing).

Data analysis

We report our findings according to STrengthening the Reporting of OBservational studies in Epidemiology (STROBE) guidance (Supplemental Material) (STROBE Group, 2007).

Statistical analyses were conducted in R Version 4.0.2 (The R Foundation for Statistical Computing, Vienna, Austria). Code for all analyses is available as Supplemental Material. Continuous data are reported using mean (standard deviation, SD) or median (interquartile ranges, IQR) where appropriate for measures of central tendency and spread. Categorical data are reported as numbers (percentages, %). A p-value < 0.05 was considered statistically significant. Relationships between categorical variables and outcome measures are presented as univariate odds ratios with accompanying p-values (Pearson’s Chi-square test with Yates’ continuity correction). A planned analysis of free-text entries in ‘Other Comments’ will be reported in a separate article.

To identify risk factors for COVID-19 amongst healthcare workers, we modelled the association between covariates chosen a priori and the COVID-19 composite endpoint using univariable and multivariable logistic regression modelling. Covariates included in the multivariable model were: age, sex, ethnicity, household composition, country of residence, main healthcare facility of work, employment role group, use of public transport to travel to work, regularity of clinical contact with suspected or confirmed COVID-19 patients, regularity of exposure to AGP(s) performed in suspected or confirmed COVID-19 patients, whether the participant had sufficient training in PPE use, whether the participant lacked access to PPE items for clinical contact with suspected or confirmed COVID-19 patients, degree of clinical contact with patients without adequate PPE, whether the participant reused disposable PPE, and whether the participant used improvised PPE. As questions on comorbidities and tobacco smoking were optional in the survey, those participants who did not answer these questions were identified as ‘Prefer not to say’. For those participants who answered ‘No’ to regular clinical contact with suspected or confirmed COVID-19 cases without adequate PPE, they were regarded as having a frequency of ‘None’ for clinical contact without adequate PPE. Findings from the regression analysis are reported as adjusted odds ratios (OR) with 95% confidence intervals (95% CI) and accompanying p-values. Quality of the final model was assessed by the Akaike information criterion (AIC) and Area Under the Receiver Operating Characteristic curve (AUROC).

Three further post hoc analyses were conducted to test the robustness of our findings:

(1) modelling was repeated in a subgroup of participants who had regular clinical contact with suspected or confirmed COVID-19 patients. This was done to test if certain workplace environments exposed participants to a greater risk of SARS-CoV-2 infection.

(2) modelling was repeated in a subgroup reporting regular exposure to AGPs conducted in suspected or confirmed COVID-19 patients. This was done to assess if participant exposure to certain AGPs were more likely to result in infection with SARS-CoV-2.

(3) a separate multivariable model was constructed using a more conservative dependent outcome variable of laboratory-confirmed SARS-CoV-2 infection, adjusting for the same covariates as in the full model above. This was done to limit the outcome to only laboratory-confirmed cases of SARS-CoV-2, thereby reducing bias from suspected cases (e.g. those who self-isolated but were never tested).

Patient and public involvement

As the survey was designed by healthcare workers, and the target population was healthcare workers, patient and public involvement was not sought.

Results

The study was conducted between 4 and 25 May 2020, and a total of 6,260 participants responded, with 6,152 eligible for analysis (Fig. 1).

Figure 1 STROBE flowchart for analysis of survey responses.

Participant characteristics

Participant characteristics for the sampled population are summarised in Table 1.

Table 1 Summary of participant characteristics, stratified by COVID-19 outcome.

	All1
(n = 6152)	COVID-19 composite endpoint2
(n = 1,806)	Self-isolated3
(n = 1,776)	Hospitalised4
(n = 49)	Lab-confirmed COVID-195
(n = 459)	
Age						
years, mean (SD)	43.2 (10.6)	41.9 (10.2)	41.9 (10.2)	42.6 (11.9)	42.5 (10.2)	
Sex						
Female	4,789 (77.8%)	1,416 (78.4%)	1,397 (78.7%)	32 (65.3%)	333 (72.5%)	
Male	1,363 (22.2%)	390 (21.6%)	379 (21.3%)	17 (34.7%)	126 (27.5%)	
Ethnic group						
Asian or Asian British	846 (13.8%)	267 (14.8%)	259 (14.6%)	12 (24.5%)	74 (16.1%)	
Black, African, Black British or Caribbean	299 (4.9%)	100 (5.5%)	98 (5.5%)	5 (10.2%)	23 (5.0%)	
Mixed or multiple ethnic groups	149 (2.4%)	48 (2.7%)	48 (2.7%)	2 (4.1%)	14 (3.1%)	
White	4,667 (75.9%)	1,330 (73.6%)	1,313 (73.9%)	29 (59.2%)	331 (72.1%)	
Another ethnic group	162 (2.6%)	53 (2.9%)	50 (2.8%)	1 (2.0%)	16 (3.5%)	
Prefer not to say	29 (0.5%)	8 (0.4%)	8 (0.5%)	0 (0.0%)	1 (0.2%)	
Household - Persons						
Lives alone	675 (11.0%)	205 (11.4%)	200 (11.3%)	6 (12.2%)	49 (10.7%)	
Lives with 1 or more persons	5,477 (89.0%)	1,601 (88.6%)	1,576 (88.7%)	43 (87.8%)	410 (89.3%)	
Household - Children						
No children	2,338 (38.0%)	664 (36.8%)	654 (36.8%)	25 (51.0%)	168 (36.6%)	
Has children	3,139 (51.0%)	937 (51.9%)	922 (51.9%)	18 (36.7%)	242 (52.7%)	
Comorbidities						
Hypertension	537 (8.7%)	158 (8.7%)	157 (8.8%)	6 (12.2%)	35 (7.6%)	
Diabetes	188 (3.1%)	57 (3.2%)	54 (3.0%)	6 (12.2%)	17 (3.7%)	
Cancer	78 (1.3%)	28 (1.6%)	28 (1.6%)	0 (0.0%)	5 (1.1%)	
Heart disease	76 (1.2%)	23 (1.3%)	23 (1.3%)	0 (0.0%)	4 (0.9%)	
Immunosuppression	109 (1.8%)	27 (1.5%)	27 (1.5%)	2 (4.1%)	2 (0.4%)	
Respiratory disease	569 (9.2%)	198 (11.0%)	192 (10.8%)	15 (30.6%)	49 (10.7%)	
Renal disease	35 (0.6%)	10 (0.6%)	10 (0.6%)	2 (4.1%)	3 (0.7%)	
Liver disease	30 (0.5%)	10 (0.6%)	10 (0.6%)	0 (0.0%)	2 (0.4%)	
Neurological disease	64 (1.0%)	18 (1.0%)	18 (1.0%)	0 (0.0%)	3 (0.7%)	
Obesity	692 (11.2%)	236 (13.1%)	234 (13.2%)	8 (16.3%)	47 (10.2%)	
None of the above	4,200 (68.3%)	1,192 (66.0%)	1,173 (66.0%)	23 (46.9%)	317 (69.1%)	
Prefer not to say	97 (1.6%)	26 (1.4%)	25 (1.4%)	0 (0.0%)	8 (1.7%)	
Tobacco smoking status						
Current or Ex-smoker within 1 year	551 (9.0%)	142 (7.9%)	139 (7.8%)	3 (6.1%)	27 (5.9%)	
Ex-smoker > 1 year	1,221 (19.8%)	361 (20.0%)	357 (20.1%)	7 (14.3%)	96 (20.9%)	
Never smoked	4,305 (70.0%)	1,279 (70.8%)	1,258 (70.8%)	38 (77.6%)	327 (71.2%)	
Prefer not to say	75 (1.2%)	24 (1.3%)	22 (1.2%)	1 (2.0%)	9 (2.0%)	
Notes:

n (%) or mean (SD).

1 All participants.

2 Participants with the presence of the COVID-19 composite endpoint.

3 Participants who self-isolated due to symptoms and/or testing positive for SARS-CoV-2.

4 Participants who were hospitalised due to suspected/confirmed COVID-19.

5 Participants who have tested positive for SARS-CoV-2.

Work details

A total of 5,518 participants were healthcare workers based in England (89.7%), followed by 321 (5.2%) in Scotland, 213 (3.5%) in Wales and 100 (1.6%) in Northern Ireland. Based on region, 1,770 (28.8%) responses were received from Greater London, with all other regions contributing less 5.5% each. Figure 2 shows responses stratified by main healthcare facility. Healthcare worker roles were grouped into doctors (1,770 (28.8%)), nurses, midwives and associated staff (2,516 (40.9%)), dentists and dental staff (198 (3.2%)), AHPs (1,118 (18.2%)), and Other (550 (8.9%)). Tables S1–S4 summarises responses into healthcare workers roles and grades (Supplemental Material).

Figure 2 Bar graph of main location of work grouped by all participants (n = 6,152) and participants where the COVID-19 composite endpoint was reported (n = 1,806).

A total of 3,902 (63.4%) healthcare workers reported regular clinical contact with suspected or confirmed COVID-19 patients. Of all participants, 2,296 (37.3%) responded as having had regular exposure to AGPs performed in suspected or confirmed COVID-19 patients. Data for areas of clinical contact and the AGPs that participants were exposed to are summarised in Figs. S1 and S2, respectively (Supplemental Material).

COVID-19 status

A total of 1,776 participants (28.9%) self-isolated because of COVID-19 symptoms (Fig. S3; Supplemental Material) or testing positive for SARS-CoV-2. Of those who self-isolated, 840 (47.3%) self-isolated for 1–7 days, 708 (39.9%) for 8–14 days, and 228 (12.8%) for more than 14 days. The total number of days of self-isolation in this cohort was between 11,870 and 21,158 days. The mean (SD) duration of self-isolation for participants who self-isolated for more than 14 days was 23.4 (8.8) days. In addition, 228 participants (12.8%) self-isolated more than once. Forty-nine (0.8%) participants were hospitalised for suspected or confirmed COVID-19. Responses for testing for SARS-CoV-2 revealed a total of 1,407 (22.9%) participants who were tested during the period covered by the survey: 948 (15.4%) had never tested positive or were awaiting test results, 20 (0.3%) were positive on blood testing and 439 (7.1%) were positive on oral/nasal swab testing.

Personal protective equipment

With regards to PPE, 4,334 (70.4%) participants answered that they had received sufficient training in the use of PPE. Throughout the timeframe of interest, 1,382 (22.5%) participants had been in a situation where they lacked access to items of PPE when having clinical contact with suspected or confirmed COVID-19 patients. Figure 3 summarises the PPE items that participants reported lack of access to under these situations. Furthermore, 1,306 (21.2%) participants had been in clinical contact with suspected or confirmed COVID-19 patients without adequate PPE. The top three reasons for these encounters without adequate PPE were ‘Patient not suspected/confirmed’, ‘Lack of PPE availability’, and ‘Senior instruction’ (summarised in Fig. S4 of Supplemental Material).

Figure 3 PPE items that participants lacked access to of those who have been in a situation where they lacked access during clinical contact with suspected/confirmed COVID-19 patients (n = 1,382).

PPE, Personal Protective Equipment; PAPR, Powered air-purifying respirator.

Univariable and multivariable modelling against the COVID-19 composite

The results from univariable and multivariable analyses of covariates from the survey and the presence of the COVID-19 composite endpoint are summarised in Table 2 and Table S5 (Supplemental Material).

Table 2 Variables and association with the composite endpoint.

	Univariable
OR (95% CI)	p-value	Multivariable adjusted OR (95% CI)	p-value	
Age					
Age (per year)*	0.98 [0.97–0.99]	<0.001	0.98 [0.98–0.99]	<0.001	
Sex					
Female	Ref	Ref	Ref	Ref	
Male	0.95 [0.84–1.09]	0.516	0.92 [0.79–1.06]	0.248	
Ethnicity					
White	Ref	Ref	Ref	Ref	
BAME	1.19 [1.05–1.35]	0.008	0.98 [0.85–1.12]	0.747	
Prefer not to say	0.96 [0.42–2.16]	1.000	0.91 [0.36–2.06]	0.825	
Household—persons					
Lives alone	Ref	Ref	Ref	Ref	
Lives with 1 or more persons; no children	0.91 [0.75–1.10]	0.344	0.87 [0.72–1.06]	0.179	
Lives with 1 or more persons; has children	0.98 [0.81–1.17]	0.825	1.00 [0.83–1.21]	0.962	
Comorbidities					
Hypertension	1.00 [0.83–1.22]	1.000	1.13 [0.91–1.40]	0.258	
Diabetes	1.05 [0.76–1.44]	0.831	1.00 [0.71–1.39]	0.986	
Cancer*	1.35 [0.85–2.16]	0.249	1.66 [1.01–2.67]	0.041	
Heart disease	1.04 [0.64–1.71]	0.962	1.18 [0.69–1.96]	0.535	
Immunosuppression	0.79 [0.51–1.22]	0.340	0.83 [0.52–1.29]	0.421	
Respiratory disease*	1.32 [1.10–1.58]	0.003	1.26 [1.04–1.52]	0.015	
Renal disease	0.96 [0.46–2.01]	1.000	1.08 [0.49–2.24]	0.833	
Liver disease	1.20 [0.56–2.58]	0.781	1.18 [0.51–2.53]	0.686	
Neurological disease	0.94 [0.54–1.63]	0.937	0.88 [0.49–1.52]	0.662	
Obesity*	1.28 [1.08–1.52]	0.004	1.31 [1.10–1.56]	0.003	
Prefer not to say	0.88 [0.56–1.38]	0.657	0.94 [0.58–1.48]	0.779	
Smoking status					
Never smoked	Ref	Ref	Ref	Ref	
Current or Ex-smoker within 1 year*	0.82 [0.67–1.01]	0.062	0.79 [0.64–0.98]	0.035	
Ex-smoker (more than 1 year)	0.99 [0.86–1.14]	0.951	1.09 [0.94–1.27]	0.238	
Prefer not to say	1.11 [0.68–1.82]	0.762	1.14 [0.68–1.87]	0.606	
Country					
England	Ref	Ref	Ref	Ref	
Northern Ireland*	0.42 [0.24–0.72]	0.002	0.44 [0.24–0.75]	0.004	
Scotland	0.90 [0.70–1.16]	0.464	0.95 [0.73–1.23]	0.702	
Wales	0.86 [0.63–1.17]	0.379	1.17 [0.84–1.62]	0.344	
Main healthcare facility					
Hospital	Ref	Ref	Ref	Ref	
Community healthcare facility	0.94 [0.81–1.08]	0.410	0.99 [0.84–1.17]	0.940	
Social care facility	0.95 [0.67–1.33]	0.819	1.17 [0.81–1.68]	0.399	
Other	0.58 [0.45–0.74]	<0.001	0.82 [0.61–1.08]	0.168	
Role group					
Nurses, midwives and associated staff	Ref	Ref	Ref	Ref	
Allied health professionals*	0.77 [0.66–0.91]	0.002	0.81 [0.69–0.96]	0.015	
Dentists and dental staff*	0.40 [0.27–0.59]	<0.001	0.52 [0.33–0.82]	0.006	
Doctors*	1.16 [1.02–1.33]	0.025	1.2 [1.04–1.39]	0.015	
Other	0.84 [0.68–1.03]	0.102	0.99 [0.78–1.24]	0.915	
Used public transport to travel to work*	1.43 [1.26–1.63]	<0.001	1.38 [1.20–1.59]	<0.001	
Regular clinical contact with suspected or confirmed COVID-19 patients*	1.52 [1.35–1.71]	<0.001	1.33 [1.15–1.54]	<0.001	
Regular exposure to AGP(s) performed in suspected or confirmed COVID-19 patients*	0.97 [0.86–1.08]	0.582	0.81 [0.71–0.93]	0.004	
Sufficient training in PPE use*	0.79 [0.70–0.89]	<0.001	0.85 [0.75–0.98]	0.023	
Lacked access to PPE items for clinical contact with suspected or confirmed COVID-19 patients*	1.74 [1.53–1.97]	<0.001	1.28 [1.09–1.51]	0.002	
Clinical contact without adequate PPE					
Never	Ref	Ref	Ref	Ref	
Rarely	1.35 [1.13–1.62]	0.001	1.06 [0.87–1.29]	0.547	
Sometimes*	2.32 [1.92–2.79]	<0.001	1.7 [1.37–2.10]	<0.001	
Often*	2.56 [1.83–3.58]	<0.001	1.84 [1.28–2.63]	0.001	
Always*	3.65 [2.18–6.10]	<0.001	2.93 [1.72–5.06]	<0.001	
Reused disposable PPE	1.21 [1.07–1.37]	0.002	0.98 [0.86–1.13]	0.821	
Used improvised PPE*	1.10 [0.94–1.28]	0.271	0.81 [0.68–0.97]	0.020	
Notes:

Univariate and multivariate odds ratio (OR), 95% confidence intervals (95% CI).

Akaike information criterion (AIC) for above model = 7,233.562; AIC for null model = 7,449.750. Area Under the Receiver Operating Characteristics (AUROC) for above model = 0.63; AUROC for null model = 0.50.

* p-value < 0.05 for multivariable model.

Ref, Reference value; BAME, Black, Asian and Minority Ethnic; AGP, Aerosol-Generating Procedures; PPE, Personal Protective Equipment.

No difference in the presence of the COVID-19 composite endpoint was seen between different ethnic groups. This persisted with constituent ethnic groups replacing the collective BAME group: Asian or Asian British (adjusted OR 0.96 (0.81–1.14), p-value = 0.643), Black, African, Black British or Caribbean (adjusted OR 1.03 (0.79–1.33), p-value = 0.845), Mixed or Multiple ethnic groups (adjusted OR 0.99 (0.69–1.40), p-value = 0.937), and Another ethnic group (adjusted OR 0.96 (0.67–1.36), p-value = 0.828) (Table S6; Supplemental Material).

Being a current or ex-smoker (within 1 year) was associated with a significant decrease in odds for the COVID-19 composite endpoint compared to participants who never smoked. To assess if this effect was due to collider bias following adjustments for comorbidities in our model, an additional model was constructed with comorbidities removed (Table S7; Supplementary Material). However, in this model, being a current or ex-smoker (within 1 year) still had reduced odds for the presence of the composite endpoint (adjusted OR 0.79 (0.64–0.98), p-value = 0.034).

Post hoc analyses

In a subgroup of participants who had regular clinical contact with suspected or confirmed COVID-19 patients (3,902 (63%)), those working in higher risk areas as defined by PHE (Public Health England, 2020c) made up 81.7%. Working in an inpatient clinic area was associated with a significant increased risk of reporting the primary endpoint (adjusted OR 1.41 (1.01–1.97), p = 0.043). The following areas were associated with a significant decreased risk: home visits (adjusted OR 0.68 (0.47–0.98), p = 0.040), ICU/HDU (adjusted OR 0.78 (0.65–0.94), p = 0.007), operating theatre (adjusted OR 0.71 (0.57–0.87), p = 0.001), radiology (adjusted OR 0.62 (0.42–0.91), p = 0.016) and other areas (adjusted OR 0.69 (0.48–0.98), p = 0.044). Table S8 (Supplemental Material) summarises the model for this subgroup analysis.

In terms of exposure to AGPs, a second subgroup analysis of participants who had been regularly exposed to AGPs used in suspected or confirmed COVID-19 patients (2,296 (37.3%)) showed that 95.8% of this cohort were exposed to procedures considered by PHE to be AGPs (Public Health England, 2020c) (i.e. this excludes ‘CPR’, ‘nebulisers’ and ‘other’ as AGPs during the study period). In this subgroup, no particular AGP was associated with a significant change in risk on multivariate analysis (Table S9; Supplemental Material).

Finally, an additional multivariable regression model constructed using a more restrictive outcome of laboratory-confirmed SARS-CoV-2 (Table S10; Supplemental Material). In this additional model, the associations between the outcome and the following factors remained significant, adjusting for other covariates: regular clinical contact with COVID-19 patients, regular exposure to AGP(s) performed in COVID-19 patients, lack of access to PPE items for clinical contact with COVID-19 patients, clinical contact without adequate PPE, and use of improvised PPE.

Discussion

Principal findings

We describe the characteristics of a sample of UK-based healthcare workers working during the COVID-19 pandemic and relate their experiences to the development of COVID-19 infection-related outcomes. The overall prevalence of the primary composite endpoint amongst healthcare workers was 29.4% over the period from 1st February to 25th May 2020. We report a substantial number of working days lost from self-isolation due to symptoms and estimate between 11,800 and 21,100 working days lost during the study period, translating to between 71 and 127 working days lost per 1,000 working days (assuming a 40-hour work week per healthcare workers). Under a quarter of participants were tested for SARS-CoV-2 throughout the period. Additionally, a number of risk factors were explored using regression modelling of the survey responses. Finally, we report that 22.5% of participants had encountered a situation where they lacked PPE items and identified a variety of PPE items that were not available.

Interpretation in the context of wider literature

The prevalence of suspected or confirmed COVID-19 amongst healthcare workers is higher in our sampled population compared to other sources (Office for National Statistics, 2020; Keeley et al., 2020; Hunter et al., 2020; El-Boghdadly et al., 2020; Pollán et al., 2020). Previous estimates have ranged from as low as 1.73% through a population survey-based approach (Office for National Statistics, 2020) to as high as between 7.7% and 24.4% via testing of healthcare staff (Keeley et al., 2020; Hunter et al., 2020; Pollán et al., 2020; Shields et al., 2020). A study amongst healthcare workers involved in tracheal intubation using a similar primary endpoint reported an overall incidence of 10.7% over a median follow-up period of 32 days (El-Boghdadly et al., 2020). The use of a composite endpoint facilitated capture of outcomes from individuals who were plausibly at risk of testing positive for SARS-CoV-2, but who were never tested. Indeed, 77.2% of our total sampled population (n = 4,745) had never been tested. During the time period, healthcare workers had to self-isolate based on clinical symptoms alone due to lack of mass testing (Dunn et al., 2020). Over three quarters of our study cohort were not tested for SARS-CoV-2. Consequently, a substantial number of working days were lost as workers had to self-isolate for prespecified durations, rather than potentially testing negative for the virus and returning to work earlier. Drawing definitive prevalence conclusions from the data reported herein is challenging due to the self-reported nature of study conduct, but the magnitude of the reported prevalence and working days lost cannot be ignored.

In our explanatory model, the presence of the COVID-19 composite endpoint was less likely in current tobacco smokers which persisted when adjustments for comorbidities were removed from the model. Petrilli et al. (2020) noted a similar protective effect against hospitalisation from COVID-19 amongst smokers in the general population, though it was reasoned that this could be due to absence of data. In a separate observational study, Williamson et al. (2020) noted that being a current smoker was associated with a lower risk of mortality after adjusting for comorbidities, which were largely driven by the adjustment for chronic respiratory disease and may also explain the mechanism behind our findings with smoking in this study. Nicotine has been posited as a potential treatment option for COVID-19 patients (Farsalinos et al., 2020). In contrast, smoking has been implicated in progression of COVID-19 infection, with recommendations for current smokers to engage with cessation (World Health Organization, 2020c; Zyl-Smit, Richards & Leone, 2020). Based on our survey data collection method, we may have missed a cohort of healthcare workers who smoke and been affected with more severe COVID-19 and who were thus, unable to participate in the survey.

Our data did not suggest any difference between White and BAME groups within the healthcare workers population for developing the COVID-19 composite endpoint, after adjusting for comorbidities (including obesity). However, amongst the hospitalised group, there was a higher proportion of BAME healthcare workers compared with the total sampled population (40.8% vs. 23.7%), particularly ‘Asian or Asian British’, and ‘Black, African, Black British or Caribbean’. Increased focus on the BAME community has resulted from findings of more severe COVID-19 infection amongst individuals of BAME origin (Intensive Care National Audit & Research Centre, 2020; Aldridge et al., 2020; Kursumovic, Lennane & Cook, 2020). PHE have previously reported on the disparities in risks and outcomes for COVID-19 infection, identifying a higher prevalence of positive tests for SARS-CoV-2 and more severe disease amongst BAME groups within the UK, though the effects of occupation and comorbidities (including obesity) were unaccounted for (Public Health England, 2020a, 2020b). Thus, despite our finding of similar risks in terms of developing the COVID-19 composite endpoint (and, therefore, possible COVID-19 infection), it remains the case that healthcare workers from BAME origins may be at risk of more severe disease and death.

We identified use of public transport to get to work as an independent risk factor for COVID-19 infection. Previous data in China has shown an association between use of public transport and spread of COVID-19 (Zheng et al., 2020; Zhao et al., 2020). Prior to the lockdown decision in the UK on the 23rd of March (Dunn et al., 2020) public transport was still operational. After this date, key workers were still allowed to utilise public transport, which included healthcare workers. Due to the presumed method of COVID-19 transmission (Public Health England, 2020d), the close proximity of individuals using public transport was likely a factor in increasing risk. However, there are many other possible confounding factors that impact on this finding which would be difficult to control for.

Adequate training and correct use of PPE (particularly during donning and doffing) are important in reducing the risk of transmission of respiratory infectious disease from patient to healthcare workers (Verbeek et al., 2020; Cook, 2020) and this was reflected in our results. This may also explain our finding that healthcare workers exposed to regular AGPs in suspected or confirmed COVD-19 patients were less likely to have the presence of the primary endpoint. Given the importance of PPE use to protect against viral transmission (Verbeek et al., 2020; Liu et al., 2020; Tabah et al., 2020), it is unsurprising that participants who lacked access to PPE items, and those who were more frequently exposed to suspected or confirmed cases of COVID-19 without adequate PPE had a higher risk of the presence of the COVID-19 composite endpoint. That nearly a quarter of UK healthcare workers reported being in such a situation is notable.

Strengths and limitations

Strengths of our study include a relatively large sample size and the inclusion of healthcare workers from all backgrounds and work environments to facilitate risk comparisons using a standardised survey. We captured granular information that has otherwise been poorly reported in prevalence studies in healthcare workers. For example, medical history and details regarding the use, or lack thereof, of PPE have not been elsewhere reported. We did not limit our recruitment to primary or secondary care; further, our sample demographics are comparable to the overall population characteristics of all healthcare workers across the NHS, which show a similar distribution by sex and ethnicity (GOV.UK, 2020; The King’s Fund, 2021) and was well-represented by a wide range of healthcare worker roles. Our findings are thus likely to be generalisable across the UK.

Several limitations need to be addressed. First, data were gathered using a survey-based approach which risks selection and recall bias. We also could not capture data from healthcare workers who died from COVID-19 infection, or those who were too ill to respond. However, our methodology allowed us to rapidly capture both objective and subjective granular data from a large number of participants. Second, we were unable to determine a denominator to quantify a response rate for this observational study. Third, the use of a composite outcome to detect suspected or confirmed COVID-19 infection in healthcare workers may have resulted in an overestimation of prevalence. However, this definition is in keeping with that used in other studies (El-Boghdadly et al., 2020) and internationally (CDC, 2020). Availability of testing for healthcare workers was also limited during early phases of the pandemic, which improved substantially as the pandemic progressed, and thus clinical diagnoses were often relied upon. On the other hand, data have estimated that 7% of healthcare workers are asymptomatic seroconverters (Office for National Statistics, 2020; Treibel et al., 2020) and thus our data could potentially represent an underestimation of COVID-19 transmission during the height of the first surge of the pandemic in the UK. Fourth, we sought some subjective data, although this was a pragmatic decision to maximise detail in responses. Fifth, several changes to national guidance and policies were made throughout the study period (Dunn et al., 2020) which may confound responses regarding PPE. Finally, all data herein are subjective and represent hypothesis-generating associations in the responding participants; further objective analyses are required.

Conclusions

We found a reported prevalence estimate of suspected or confirmed COVID-19 infection of nearly a third, based on a COVID-19 composite endpoint, amongst healthcare workers within the UK. As a consequence of self-isolation, between 11,000 and 21,000 days of clinical service was lost which could have been mitigated by more robust SARS-CoV-2 testing strategies. We also present several risk factors associated with reporting of this endpoint, lack of PPE being an important consideration. Though global vaccination programs are being rolled out, policymakers still need to ensure adequate PPE supplies to all healthcare workers in preparation for future surges in COVID-19 cases and that accessible, rapid, accurate testing strategies are available to improve healthcare workforce planning and continuation.

Supplemental Information

Supplemental Information 1 STROBE Checklist for healthcareCOVID.

Click here for additional data file.

Supplemental Information 2 Supplemental Material for healthcareCOVID.

Contains: (1) overview of healthcareCOVID survey, (2) STROBE checklist, (3) Disclaimer for healthcareCOVID survey, and (4) eFigures 1 - 4 and eTables 1 - 10.

Click here for additional data file.

Supplemental Information 3 Raw Dataset (Deidentified) with R Code.

Raw dataset from participant survey responses (deidentified) with accompanying R code used for statistical analyses (labelled in running order).

Click here for additional data file.

Supplemental Information 4 healthcareCOVID Service Evaluation Protocol.

Click here for additional data file.

The authors would like to thank all of the 6,260 healthcare workers who participated in the healthcareCOVID study. In addition, we would like to thank Steve Palmer and Knack (Pennsylvania, USA) for providing assistance with the web-based software, database and server space used for the healthcareCOVID study.

Additional Information and Declarations

Competing Interests

Author Contributions

Human Ethics

Data Availability

The authors declare that they have no competing interests.

Justin Kua conceived and designed the experiments, performed the experiments, analysed the data, prepared figures and/or tables, authored or reviewed drafts of the paper, and approved the final draft.

Reshma Patel performed the experiments, authored or reviewed drafts of the paper, and approved the final draft.

Eveliina Nurmi performed the experiments, authored or reviewed drafts of the paper, and approved the final draft.

Sarah Tian performed the experiments, authored or reviewed drafts of the paper, and approved the final draft.

Harpreet Gill performed the experiments, authored or reviewed drafts of the paper, and approved the final draft.

Danny J.N. Wong performed the experiments, analysed the data, authored or reviewed drafts of the paper, and approved the final draft.

Calvin Moorley performed the experiments, authored or reviewed drafts of the paper, and approved the final draft.

Dmitri Nepogodiev analysed the data, authored or reviewed drafts of the paper, and approved the final draft.

Imran Ahmad performed the experiments, authored or reviewed drafts of the paper, and approved the final draft.

Kariem El-Boghdadly conceived and designed the experiments, performed the experiments, analysed the data, authored or reviewed drafts of the paper, and approved the final draft.

The following information was supplied relating to ethical approvals (i.e., approving body and any reference numbers):

The study was prospectively registered as a service evaluation project at Guy’s and St Thomas’ NHS Foundation Trust (Service Evaluation ID: 10834) and was deemed to not require ethical approval by the Research and Development Department and the Health Research Authority Decision Tool.

The following information was supplied regarding data availability:

Raw data and code are available in the Supplemental Files.

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
