# Peer review of "healthcareCOVID: a national cross-sectional observational study identifying risk factors for developing suspected or confirmed COVID-19 in UK healthcare workers"

_PeerJ, doi:10.7717/peerj.10891_

## Round 0.1 · original submission · Major Revisions

The reviewers have raised substantial interest in this work and have found the study well conducted. However, they have also raised some concerns about the conclusions from the meta-analysis that should be addressed before the manuscript can be reevaluated for publication.

Reviewer 1 ·

Basic reporting

Excellent

Experimental design

See below.

Validity of the findings

I think, overall, this is a well conducted study that provides potentially useful observations regarding health care worker risk.

There are considerable challenges in conducting epidemiological research of this nature when the clinical performance of different PCR and serological diagnostics remain unclear and their availability limited.

However, the use of the composite outcome concerns me considerably (see below). Until a sample of individuals who have definitively suffered SARS-CoV-2 infection (PCR/serology) has been captured and the uncertainty within those diagnostic tools known, understanding the risk factors for COVID-19 is challenging.

I think these limitations need further acknowledgement, even if they can't be addressed.

Additional comments

The authors have undertaken a large survey based study of UK healthcare professionals to identify risk factors for potential of confirmed COVID-19 and provide helpful descriptive data from multiple healthcare domains including the number of days isolated, PPE training, provision and use.

The data is well presented and the methodology clear.

My only significant criticism of the study design is on the use of the composite outcome (see below).


Major points

With respect to the authors’ primary outcome, I believe that there are significant limitations with the inclusion of “suspected” COVID-19 cases in the analysis presented. 

Using reportedly sensitive serological assays, the seroprevalence in symptomatic UK health care workers is around 30%. (Shields et al - Thorax doi: 10.1136/thoraxjnl-2020-215414), or using a composite PCR + equally sensitive flow cytometric-based serological assay, around 50% (Houlihan et al - Lancet doi.org/10.1016/S0140-6736(20)31484-7).

This suggests that many (perhaps the majority) of individuals did not isolate COVID-19. It is therefore likely that this study has enrolled considerable numbers (maybe more than 50%) of HCW who did not have COVID-19. Equally, the authors cite studies estimating up to 10% of individuals asymptomatically seroconvert - estimates from other studies are greater than this (e.g. Hains et al JAMA. 2020 Jun 16; 323(23): 2424–2425, Shields et al - Thorax doi: 10.1136/thoraxjnl-2020-215414, Eyre et al - eLife 2020;9:e60675)

It is of great importance that the occupational risk factors for COVID-19 in health care workers are better understood so countermeasures can be employed to reduce that risk. Although the authors acknowledge this issue (line 345), there is no consideration or quantification of how much of an impact this has on their analysis; it is likely to be considerable if 50% didn’t have disease in question.

The justification of the approach by comparison to other studies (line 348) is not particularly constructive.

Perhaps a more helpful piece of analysis is to only consider the PCR proven or antibody proven disease (n=459) within a logistic regression models.

2. There are a couple of interesting risk factors that merit further discussion - firstly - the authors show that there are non-hospital risk factors for their composite outcome (e.g. travel to work by public transport). This strongly suggests that not all COVID-19 in HCW is health care related, which is worth discussing further.

3. Related to above, the other significant limitation is that the survey distributed did not document when people had COVID-19. National data suggests that there is a temporal delay in the peak of community COVID and hospital admissions. The survey has collected data on exposure to infectious patients, but this may have occurred after their COVID-19 illness.

4. Current smoking emerges as a protective factor in multiple models - I have not seen this observation documented before and perhaps it could be discussed further.


Minor points

Line 66 - the authors may wish to consider the work by Shah et al - BMJ 2020; 371 in relation to health care worker risk. In my opinion this provides more robust evidence of the burden of COVID-19 in HCW than the Heneghan piece currently cited.

Line 69 - the statement that health care workers are exposed to “higher viral loads” is conjecture. Indeed, Ra et al, Thorax 2020 http://dx.doi.org/10.1136/thoraxjnl-2020-215042 provides evidence of equivalent viral load - determined by PCR CT - in asymptomatic and symptomatic individuals.

Line 84 - the authors should clarify specifically what they mean by “the nature of SARS-CoV-2 transmission”; aerosolised, droplet transmission

Line 127 - Survey administration: I think the organisations that distributed the survey on behalf of the authors should be directly included in the methods sections, rather than in the appendix. It is also not clear what the reach of the survey was - I am unable to find how many people this survey was directly distributed to (e.g. via mailing lists). If the survey was publicly available, how did the authors confirm the respondents were health care professionals?

Line 317 - “Thus, despite similar susceptibilities to the disease” - not clear what you mean here. Are you proposing similar exposure to the virus, similar chances of being infected by the virus (which is not proven) or similar susceptibilities of infection becoming symptomatic illness?

eTable 9 - sufficient training in PPE has an Asterix but isn’t statistically significant. There are similar occurrences in other tables, which should be reviewed by the authors.

·

Basic reporting

The paper is very clearly written, well motivated, and well executed. The raw data has been made available. Some very minor comments:

1. Do you mean 'another ethnicity' rather than 'another ethnic' on line 137?
2. Lines 151-155 - I think you should reiterate here that there was a free-text gender identity option.

Experimental design

1. Your finding that smoking reduces the risk of the composite endpoint is suprising, and not in line with other findings e.g:

The effect of smoking on COVID‐19 severity: A systematic review and meta‐analysis, Reddy et al (2020)

Is there a smoker’s paradox in COVID-19? Shariq Usman et al (2020)

I appreciate the efforts to understand if this is induced by collider bias, but think you could go further. Can you see if the association holds in hospitalised patients? If you still find this result then it warrants more coverage in your discussion, including putting the finding into context with the rest of the literature.

2. I can't find mention of what is adjusted for in the multivariable models, please clarify.

3. There is mention of model validation using the AIC and AUROC, but these are not presented in the results. Either these results should be included, or this claim removed (lines 183-185),

Validity of the findings

1. It would be good to comment on how the respondent population compares to the UK population of healthcare workers, in order to understand how generalisable your findings may be. This also touches on point 2 of Experimental Design - without knowing what you control for in your multivariable models, it is difficult to assess whether your findings could be skewed by over/under representation of certain groups.

2. Lines 310 - is the difference in BAME hospitalised workers statistically significant?

---

## Round 0.2 · accepted · Accept

Congratulations on successful revision and acceptance of your paper.

Reviewer 1 ·

Basic reporting

Excellent

Experimental design

Previous concerns addressed

Validity of the findings

Well described, limitations acknowledged and an important contribution.

Additional comments

Thank you for the opportunity to review the submission following revisions.

My concerns have been addressed, in particular with respect to the additional analysis performed on the laboratory confirmed COVID-19 cases. The study is well conducted and the strengths and limitations acknowledged. I congratulate the authors on their tremendous efforts in producing this work which meaningfully contributes to the understanding of healthcare associated risk during a very challenging time for the NHS.

·

Basic reporting

I'm satisfied with the authors' response to my comments.

Experimental design

No comment.

Validity of the findings

No comment.